# Methylene Blue—Current Knowledge, Fluorescent Properties, and Its Future Use

**DOI:** 10.3390/jcm9113538

**Published:** 2020-11-02

**Authors:** Tomasz Cwalinski, Wojciech Polom, Luigi Marano, Giandomenico Roviello, Alberto D’Angelo, Natalia Cwalina, Marcin Matuszewski, Franco Roviello, Janusz Jaskiewicz, Karol Polom

**Affiliations:** 1Department of Surgical Oncology, Medical University of Gdansk, 80-214 Gdansk, Poland; Cwalinski.tomasz@gmail.com (T.C.); janusz.jaskiewicz@gumed.edu.pl (J.J.); 2Department of Urology, Medical University of Gdansk, 80-214 Gdansk, Poland; wojtek.polom@gmail.com (W.P.); marcin.matuszewski@gumed.edu.pl (M.M.); 3Department of Medicine, Surgery and Neurosciences, Unit of General Surgery and Surgical Oncology, University of Siena, 53-100 Siena, Italy; luigi.marano@unisi.it (L.M.); franco.roviello@gmail.com (F.R.); 4Department of Health Sciences, University of Florence, viale Pieraccini 6, 50139 Florence, Italy; giandomenicoroviello@hotmail.it; 5Department of Biology and Biochemistry, University of Bath, Bath BA2 7AY, UK; ada43@bath.ac.uk; 6Department of Pediatrics Ascension St. John Children’s Hospital, Detroit, MI 48236, USA; ncwalina@gmail.com

**Keywords:** methylene blue, fluorescence, fluorophore

## Abstract

Methylene blue is a fluorescent dye discovered in 1876 and has since been used in different scientific fields. Only recently has methylene blue been used for intraoperative fluorescent imaging. Here, the authors review the emerging role of methylene blue, not only as a dye used in clinical practice, but also as a fluorophore in a surgical setting. We discuss the promising potential of methylene blue together with the challenges and limitations among specific surgical techniques. A literature review of PubMed and Medline was conducted based on the historical, current and future usage of methylene blue within the field of medicine. We reviewed not only the current usage of methylene blue, but we also tried to grasp its’ function as a fluorophore in five main domains. These domains include the near-infrared imaging visualization of ureters, parathyroid gland identification, pancreatic tumors imaging, detection of breast cancer tumor margins, as well as breast cancer sentinel node biopsy. Methylene blue is used in countless clinical procedures with a relatively low risk for patients. Usage of its fluorescent properties is still at an early stage and more pre-clinical, as well as clinical research, must be performed to fully understand its potentials and limitations.

## 1. Introduction

Ever since German chemist Heinrich Caro first prepared methylene blue in 1876, it has been described as the first fully synthetic drug used in medicine [1]. Throughout many decades, a multitude of usages have been discovered amongst the various fields of science including clinical medicine and surgery [2,3].

Methylene blue is commonly used in medical practice, especially as a dye in microbiological staining. This function has now lent itself to the operating theatre, where methylene blue is often used as an marker and indicator in various surgical techniques [4,5,6]. Recently, it has been used for the detection of intestinal, enterovesical, and bronchopleural fistulas [7,8]. It has been also used as a drug for treating methaemoglobin-induced encephalopathy and often aids in treating dermatological diseases, such as herpes labialis, eczema herpeticum, oral candidiasis, and cutaneous leishmaniasis [2,9,10,11,12,13,14]. Another field where methylene blue (MB) can be used is as an antidote in cyanide poisoning [14,15,16]. Historically, it was used in patients with urinary tract infection—as an antiseptic and stimulant of mucous surfaces, although it is no longer recommended in either [4,17]. Additionally, we can use MB intravenously in different doses to visualize enlarged parathyroid glands, observe the ureters and evaluate any potential damage [18,19]. Methylene blue (MB), also known as methylthioninium chloride, is a thiazine dye that can be used as a medication, and can be administered orally, subcutaneously or intravenously. The intravenous setting should be administered slowly—usually over 3 to 10 min [20,21]. Methylene blue is excreted in the urine 4 to 24 h after intravenous administration, with a half-life of 5 to 6.5 h [17]. Despite this, it is not necessary to reduce the dose in patients with renal insufficiency. However, it should be used with caution [14]. There is no specific dosage guidance for patients with liver failure [20].

Lastly, methylene blue can be used to detect lung nodules during thoracoscopic procedures [6] Interestingly enough, some patients metabolize the methylene blue molecule into a non-fluorescent leucomethylene blue with an unclear impact on the patient [17] Despite all of this, the most traditional use of methylene blue remains the detection of sentinel nodes in different cancers [22,23,24,25,26,27] as initially proposed by Giuliano et al. for breast cancer in 1994 [28].

The main effect of methylene blue on molecular level is through modulation of the cyclic guanosine monophosphate (cGMP) pathway. Additionally, multiple cellular and molecular targets of the MB compound have been identified. Many studies describe in detail the mechanisms of MB at the molecular level [19,29].

## 2. Fluorescence

Near-infrared (NIR) imaging is a novel technique that uses near-infrared light to help surgeons visualize different structures and allows them to see beyond the normal white light spectrum [30]. The NIR fluorescence visualization requires a fluorophore to work together with an imaging system to detect and measure its fluorescence after excitation. Depending on the clinical situation, fluorophores can be administered locally, but also systemically.

Fluorophores are excited by a specific wavelength of near-infrared light that reemits back an appropriate light visualization. The wavelength of this light is a longer one [31]. Due to this, near-infrared (NIR) light (700–900 nm) allows visibility of different structures about a 10 mm depth into the tissue, in the distinction of visible light [32,33,34] Within the 700–900 nm spectrum, no autofluorescence is detected in tissues. The NIR fluorescence imaging requires the fluorophore coupled with an imagining system to show the fluorescence after excitation, then the compound detection is measured afterwards [32]. Many of the NIR fluorescence systems available on the market can alter the fluorescent signal with an overlaid color, which facilitates intraoperative orientation.

In most cases, lamps in the operating theatre have to be dimmed to the same level used during laparoscopic procedures [26].

The Federal Drug Administration (FDA) and the European Medicines Agency (EMA), have approved the clinical use of Indocyanine Green (ICG), Methylene Blue (MB) and 5-amino-levulinic acid (5-ALA) fluorophores. In our review, we focus on methylene blue, emphasizing that it is not considered a pure probe, due to its visual light emission spectrum of 400–700 nm [35]. Methylene blue has an excitation peak of about 700 nm, an excitation wavelength of 668 nm, plus an emission of 688 nm, which is visible to the naked eye [35,36]. Similar results were presented by other laboratories. Methylene blue presents with an absorption peak at 665 nm with a shoulder at 610 nm, presenting an additional peak at 293 nm [37]. It has a quantum yield of 0.52. Moreover, the emission spectra of MB is about 686 nm with a stoke shift of 21 nm. The clinical properties of MB are limited by its’ hydrophobic nature. This is the reason why methylene blue presents less tissue penetration while at the same time, more autofluorescence of the background tissue. In comparison, the ICG fluorescence spectrum is about 800 nm, with 5-ALA has a spectrum of about 510 nm, outside of the NIR fluorescence range [38,39].

## 3. Fluorescent Clinical Use

Methylene Blue when used together with a special NIR system for the detection of fluorescence, allows for the visualization of previously hidden structures during surgery. Below, we describe the major domains where the MB NIR guided surgical technique can be used.

### 3.1. Visualization of Ureters

Iatrogenic ureter injury during surgery is a rare adverse event, but is considered one of the most serious complications. It is especially common during lower abdomen surgeries—especially gynecological interventions, where the prevalence ranges between 0.15 to 1.5% [36,40,41,42]. In colorectal procedures, the lower third of the left ureter is most prone to damage. [43]. Faster recognition of the ureter, although challenging in cases of previous surgery or radiotherapy of the pelvic floor, not only makes it easier to avoid potential damages, but it also helps in identifying different anatomical structures such as nerves and gonad vessels [44]. Intraoperative diagnosis of ureteral injury enables immediate repair and reduces negative consequences on patient’s health [45] Intra-operative localization of the ureter often uses a light ureter catheter, particularly in laparoscopic surgery. However, this is an additional invasive procedure for patients, with an increased risk for further complications [46].

Verbeek et al. was the first to report the use of NIR fluorescence-guided surgery to visualize the ureters by intravenous injection of methylene blue [47]. In 12 patients, both ureters were detected 10 min after peripheral vein injection, with the signal lasting up to 60 min after administration. Three different doses of MB 0.25, 0.5, and 1 mg/kg were used. Statistically, these groups differed only in the time of exposure. Conversely, Al Taher et al. managed to visualize both ureters in only 5 out of 10 patients [48].

Barnes et al. reported identification of 64 out of 69 ureters during colorectal surgery of 40 patients [36]. Moreover 14 of ureters were not visible under white light. A group of 50 ureters were visible under both white and fluorescence light, but 14 of them were easier and faster to identify using the NIR camera. From the anatomical point of view, in 10 cases ureter was seen in a different localization than expected (14.5%). The authors divided patients into four groups based on MB concentrations used: 0.25, 0.5, 0.75, or 1 mg/kg. A significant difference has been shown between the mean signal-to-background ratio across all time points. The highest mean signal-to background ratio was observed using the 0.75 mg/kg dose (mean = 5.29, SD = 2.72, 95% CI 4.84–5.75), and the lowest at 1 mg/kg (mean = 3.66, SD = 1.89, 95% CI 3.37–3.39). On the other hand, Yeung et al. showed that intra-operative concentrations of 1 mg/kg gave the strongest signal [49]. After administration of methylene blue intravenously during surgery, the highest fluorescence was obtained between 9 and 20 min after administration with a peak at 14.4 min. In all cases, the autofluorescence was low and the mean signal to the background ratio was 2.74. Figure 1 presents our experience in searching ureter during anterior resection of the rectum using Quest Spectrum camera, Quest, Netherlands (Figure 1).

One of the limitations of using methylene blue to visualize the ureters is the potential impairment of renal function because it is excreted by the kidneys. Another limitation is the restriction of MB use only for patients who are capable of converting MB into the non-fluorescing leucomethylene blue, which is caused by the reduction and/or acidity of the environment [17,50].

Allergic reactions to MB occur more frequently at doses above 5 mg/kg, therefore, use of methylene blue should be with the smallest efficacious dose [50,51,52,53]. With lower doses of MB used, it is still possible to visualize ureters with NIR fluorescence and reducing the risk of side effects at the same time [47]. The standard 1% concentration (i.e., 31.3 mmol/L) of MB used in clinical settings, such as that used during sentinel node biopsy, for instance, presents no fluorescence due to the quenching phenomenon [50]. The quenching threshold of MB diluted in urine is less than 20 μmol/L. In higher concentrations, moderately high NIR fluorescence is visible, with peak excitation at 668 nm, extinction coefficient of 69,100 mol/L^−1^ × cm^−1^, peak emission at 688 nm [50].

When diuretics are used in conjunction with methylene blue, the increase in urinary excretion does not affect the fluorescence of the MB dye. This could be related to the greater dilution of urinary-methylene blue [50]. Matsui et al. assumed that this phenomenon is caused by furosemide, which affects the conversion of methylene blue into non-fluorescent leucomethylene blue [50].

Novel experimental dyes are currently in development [50]. In the review by Slooter et al., eight experimental dyes that could be used to identify ureters were described [54]. Two of the proposed dyes present interesting characteristics: CW 800–BK and ZW 800–1. Both are currently being investigated in ongoing clinical trials and the results of those trials should be available soon. The remaining six dyes are still in the earlier stages of research: CW 800–CA, Fluorescein, Liposomal ICG, Genhance 750, UL–766, Ureter glow.

### 3.2. Thyroid and Parathyroid Glands

Localization and dissection of the parathyroid glands is still a formidable challenge during surgery. Detection of the enlarged glands is often difficult due to their variability in number and location. Incorrect diagnosis of the parathyroid glands during surgery may result in serious complications [55]. The first report of using methylene blue to visualize the parathyroid glands dates back to 1971, when after injecting a high dose of MB (5 mg/kg) intravenously, surgeons noticed that the parathyroid glands gradually changed in color (blue) for one hour, followed by a gradual change back to normal after two and a half hours [56]. Moreover, the normal parathyroid gland tissue was stained blue only at the periphery of the gland, while the adenomas completely changed color. Despite these results, there are currently no prospective clinical trials using high doses of methylene blue during surgery on the parathyroid glands [57]. This is most likely due to the increased risk of side effects associated with the higher dose of MB [57].

Currently, there are nine parathyroid identification methods available on the market: autofluorescence spectroscopy, autofluorescence imaging, ICG imaging, methylene blue fluorescence imaging, 5-ALA, optical coherence tomography, laser speckle contrast imaging, dynamic optical contrast imaging, and Raman spectroscopy [35,58,59,60,61,62,63,64,65].

Methylene blue is traditionally injected intravenously in a high dose (3–7.5 mg/kg) to allow for the naked eye to be able to see the enlarged parathyroid glands as they are stained blue, but as mentioned earlier, these doses are associated with many adverse effects and should be used with caution [66,67]. The NIR fluorescence technique makes it possible to detect the glands by using lower doses of MB. According to Hillary et al. the optimal dose for this technique is 0.4 mg/kg, which makes it possible to distinguish the parathyroid glands from the surrounding tissue for a reasonable amount of time [26]. However, directly after intravenous administration of this MB dose, there can be a temporary false decrease in hemoglobin saturation, with values reading as low as 65%. This drop disappears after about 30 s, and is due to a temporary impaired reading of the pulse oximeter—a disorder of light absorption in the blood [68]. There are no hemodynamic changes observed, and the oxygen saturation readings quickly return back to baseline [26].

The lowest dose of methylene blue that has been administered intravenously to differentiate parathyroid glands from the surrounding thyroid tissue is 0.05 mg/kg, but the fluorescence only lasts for a few seconds, which makes it less than ideal for resections of the parathyroid glands [26].

Fluorescence of an abnormal parathyroid gland is greater than a normal one [26]. However, the Mc Wade et al. study shows that there is no significant difference of near infrared fluorescence visualization between a healthy parathyroid gland (signal-to-background ratio of 4.51 ± 1.24) and an abnormal one (4.81 ± 0.80) [55]. During the injection of higher doses of MB, the fluorescence of abnormal parathyroid glands increases more compared to the surrounding tissue, especially to that of normal parathyroid glands [26].

Surprisingly, we can observe auto-fluorescence of parathyroid glands prior to fluorophore injection [26]. De Leeuw et al. [58], Mc Wade et al. [55], and Paras et al. [69] all mention this in their studies, where they detected the parathyroid glands by using their autofluorescence at wavelengths of 750 nm and 785 nm, which is 2–11 times more visible then the thyroid and surrounding structures [55,58,69]. Hillary et al. demonstrated the same phenomenon by using a wavelength of 680 nm [26]. The cause of parathyroid hyper-fluorescence is unknown and seems to be independent from its’ histological structure and intraoperative blood supply [58,70].

In search for parathyroid adenomas, Van der Vorst et al. administered 0.5 mg/kg of MB (at a concentration of 10 mg/mL) intravenously for 5 min to a group of 10 patients after dissecting the neck tissues [71]. This resulted in 9 out of 10 adenomas detected using the NIR methylene blue imaging method during the operation. Furthermore, only 7 out of 9 patients had a positive preoperative 99mTC-sestamini single photon emission computed tomography scans and in 2 of the patients the adenomas were only seen with the NIR MB visualization [71].

### 3.3. Pancreatic Neuroendocrine Tumor

Methylene Blue accumulates in neuroendocrine tumors when injected intravenously, but the precise mechanism of this action is unknown. Initial reports of its clinical application have been described [72].

The first neoplastic lesion of the pancreas (insulinoma) to be stained by using high doses (5 mg/kg) of MB was seen in 1974 [73]. Winner et al. showed that by using MB as a fluorescent dye, it can be useful in visualizing insulinoma in NIR light, in an animal model [72]. When intravenously injecting MB, the NIR fluorescence from the pancreas remains relatively at the same level of a 3.0 signal to background ratio for about 60 min. Depending on the usage of MB, a different signal to background ratio can be achieved. A higher ratio is achieved when the MB administration occurs in a rapid bolus (5–20 s), when compared to a slow infusion (15–20 min). However, both techniques of administering MB displayed contrast within the pancreas. The animal models (pigs) were divided into individual groups of test subjects and received different doses of methylene blue—0.25, 0.5, 1, and 2 mg/kg. The signal to background ratio (SBR) was ≥1 in doses greater than 1 mg/kg, and was statistically significant with the baseline (*p* < 0.05). The lack of statistical difference in SBR doses of 1 and 2 mg/kg may be associated with a plateau phase of the signal strength, thus reaching the quenching threshold or saturation of the signal absorption process. Neoplastic lesions (insulinoma) of the pancreas were able to be distinguished from healthy tissue in NIR fluorescence within 2 min after intravenous injection of MB. Furthermore, NIR fluorescence was able to detect neoplastic lesions of the pancreas with a diameter less than 1 mm. The higher uptake of MB in the neoplastic tissue showed an insulinoma to pancreas ratio of 3.7 ± 0.5.

Preoperative and intraoperative difficulties of visualization of the pancreatic neuroendocrine tumors (PNET) may lead to incomplete resection, but in the case of suspected PNET associated multiple endocrine neoplasia type 1 and 2, surgeons have been able to use NIR imaging during surgery to identify lesions [74]. In one particular case, tumors were identified using preoperative magnetic resonance imaging studies. Only four lesions were visible during positron emission tomography (PET) scans. Using near-infrared imaging with MB allowed for visualization of the neuroendocrine lesions intraoperatively, with more than 20 fluorescent lesions detected in the entire pancreas. This discovery led to a change in the surgical approach and a total pancreatectomy was performed.

Pancreatic solitary fibrous tumors have also been detected using MB as a fluorescent dye with a 1 mg/kg dose, resulting in a SBR of about 3, which remained stable for nearly 15 min [75]. As a result, it was possible to localize a tumor in the uncinate process of the pancreas by using NIR fluorescence.

The possibility of using MB in NIR visualizations of paraganglioma with a smaller second lesion not seen by the naked eye has also been identified [76].

### 3.4. Breast

Intraoperative localization of breast cancer tumors is challenging and the high rate of positive margin resections is still a vast problem [77]. Therefore, intraoperative techniques capable of visualizing cancer cells are needed. Based on an analysis of different studies, a total of 402,357 patients diagnosed with breast cancer (invasive and ductal carcinoma in situ) were identified, with a mean re-operation rate (re-excision or mastectomy) of 27.49%, varying between treatment centers [78].

An interesting application of methylene blue and its fluorescent properties is its role in breast cancer margin detection during breast-conserving treatment, as proposed by Tummers et al. [79] using NIR fluorescence imaging, they detected tumors of the breast in 20 of 24 cases [40]. Patients were divided into two groups: those with early imaging, with methylene blue administered intravenously 5 min before the start of the operation, and those with delayed imaging, with methylene blue administered 3 h prior to the surgery. All patients received a 1 mg/kg dose of MB. [79]. There was no statistical difference between the groups when analyzing the tumor to background ratio. Among these patients, four had positive surgical margins. In two cases with the use of NIR fluorescence, a positive margin was visible on the surface of the excised tissue in the post-operative bed and also in the cut margin. In one of the cases, by using MB fluorescence, a second surgery was avoided. In one patient, the tumor was not visible using the NIR fluorescence. The authors suggested that this may have been the result of the transformation of methylene blue into a non-fluorescent form. Another negative case was the result of logistical problems during the study. There were no statistical changes in the histopathological type of breast cancer (*p* = 0.22).

Another study by Zhang et al. showed the possibility of using MB fluorescence imaging to identify breast cancer [59]. This study involved 30 breast cancer patients divided into two groups—patients who received pre-operative chemotherapy, and patients without neoadjuvant treatment. All patients received a 1 mg/mL dose of methylene blue intravenously 3 h before their surgery. To aid in fluorescence evaluation, a camera was used that had been prepared for methylene blue visualization. In 16 out of 20 patients (80%) who did not receive pre-operative chemotherapy, the dissected breast specimens showed a fluorescent contrast (signal-to-background ratio: 1.94 + 0.71). In the group of patients that received neoadjuvant chemotherapy fluorescence of the tumor could be observed in 3 of 10 cases (30%), suggesting that neoadjuvant chemotherapy has an impact on the imaging result (*p* < 0.05) and that the detection of breast cancer using MB fluorescent contrast was strongly affected. It is worth mentioning that in 5 cases, the fluorescence showed additional tissue that was excised in the group of patients without neoadjuvant chemotherapy. During the histopathological examination, this additional tissue turned out to be benign hemorrhagic tissue. The statistical analysis of the 35 samples showed a sensitivity of 0.63 and a positive predictive value of 0.79 by using MB-based NIR fluorescence.

In the paper by Jiang et al., the authors presented the first cases of sentinel node biopsy using methylene blue together with fluorescence imagining [60]. These promising videos showed potential benefits on the identification of lymph vessels, location of sentinel nodes, and the patterns of breast lymphatic flow. Future studies on this topic are needed to show its real potential as well as limitations.

## 4. Side Effects

Methylene blue is a safe drug if used within therapeutic doses of <2 mg/kg [35]. Patients can experience slight pain after intravenous injections, which resolves after flushing the access site with saline [59,79]. Other adverse symptoms include a burning sensation, rash, abscess, necrosis, and even ulceration [4]. Green colored urine can even be observed after administration of MB [4]. Intravenous injection of MB causes a temporary desaturation in blood, which can be measured when using pulse oximetry, but oxygen levels return to a normal shortly thereafter [61,62,68]. High intravenous doses of methylene blue can cause nausea, vomiting, abdominal pain, precordial pain, dizziness, headache, profuse sweating, dyspnea, hypertension, and mental confusion [4,17,20]. With subcutaneous and intradermal administration, adverse skin reactions, such as erythematous macular lesions, superficial ulcers, and abscess formation may form [63,64]. In higher systemic doses, toxic reactions may occur, such as cardiac arrhythmias, coronary vasoconstriction, lower cardiac output, decreased renal function, as well as decreased mesenteric blood flow. Pulmonary vascular pressure can increase together with pulmonary vascular resistance and impaired gas exchange can be observed [35,62].

Localized skin and fat necrosis have been reported after subcutaneous injection during sentinel lymph node biopsy [63]. Tattoo persistence has also been observed up to 1 year after the operation, as well as prolonged postoperative radiotherapy and breast edema [65]. Severe immunoglobulin hypersensitive reaction has been reported in 1% MB when used for tubal permeability during general anesthesia [80,81]. According to the review by Bezu at al. [80], anaphylactic reaction to methylene blue dye during sentinel node biopsy is extremely rare. The prevalence of reactions to other blue dyes, namely isosulfan blue and patent blue V, varies between 0.07% and 2.7% [80].

Serotonin syndrome may occur after using methylene blue, especially if used together with other serotonergic drugs like selective serotonin reuptake inhibitor (SSRIs) and serotonin-norepinephrine reuptake inhibitor (SNRIs) drugs. Most cases of serotonin syndrome occurred after doses of 1–8 mg/kg MB via intravenous injection used when trying to visualize the parathyroid glands [34,82] in patients taking these medications. As such, the use of methylene blue should be avoided in these patients [10,20,82].

Methylene blue is contraindicated in patients who experienced hypersensitivity reaction and in patients with renal insufficiency because it is excreted through the kidneys [83]. It is also contraindicated in glucose-6-phosphate dehydrogenase (G6PD) deficiency and in patients with Heinz body anemia [84].

Methylene blue is potentially teratogenic and should not be used during pregnancy [4,20,85,86].

## 5. Conclusions

Methylene blue has been used in the field of medicine for decades, due to its accessibility, wide safety profile, and proved versatility. Despite its limitations, it has shown tremendous promise when used as a NIR fluorophore, especially thanks to its fluorescent properties. The field of near-infrared imagining with methylene blue as a fluorophore is an interesting surgical technique that demands further clinical investigation.

## Figures and Tables

**Figure 1 jcm-09-03538-f001:**
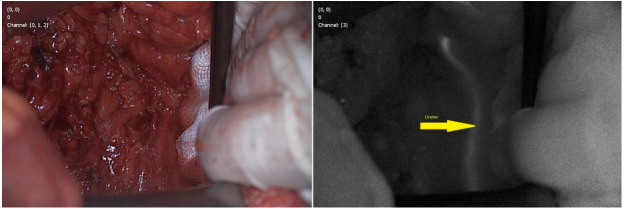
Fluorescent visualization of a ureter during rectal cancer anterior resection.

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
