# Peer review of "Methylene Blue—Current Knowledge, Fluorescent Properties, and Its Future Use"

_jcm, 2020, doi:10.3390/jcm9113538_

Round 1

Reviewer 1 Report

The review “Methylene Blue – current knowledge, fluorescent properties and its future use.” By T. Cwalinski  et al. (please note that in the author list the first name is written after the last name, i.e. Tomasz is the first name not the last name of the first author, etc.) features the use of fluorescene dyes, in particular methylene blue, in therapy and diagnostics, and provides background information to foster the future use of the dyes fluorescent properties in clinical settings.

Therefore, this review is very valuable.

 In order for the review to become even more useful,  I would suggest to reorder the Introduction and to shift the second paragraph (line 39-45) behind the sentence in line 56 ending with “ ..potential damage.”.

I further would recommend to include more spectroscopic details about methylene blue in the second section (2. Fluorescence). It would be informative to show an absorption and emission spectrum of methylene blue and to directly specify the wavelength of absorption and emission. And to indicate that it is a hydrophobic molecule.

In line 81 “an excitation peak of about 700 nm, …… an emission of  688nm… “: Please rephrase the sentence such that it is physical correct (stokes shift between absorption and emission). I would suggest to site the following reference J Pharm Anal. 2017 Feb; 7(1): 71–75 or a similar reference.

Figure 1. Is the Figure legend missing? It would be nice to highlight in the fluorescence image with an arrow the “ureter during anterior resection”

Minor point: Subheadings starting from line 121, 177,226,262 etc  need to be reformatted. I think they also need numbers, as the subheadings ended with “3. Side effects”

Author Response

Dear Editor and Editorial staff,

Please find attached the revised version of the manuscript JCM-961668, entitled: “"Methylene Blue - current knowledge, fluorescent properties and its future use."

The paper has been revised in accordance with the Reviewers’ comments. We would like to thank the Reviewers for their important comments and suggestions, which have notably improved the quality of our manuscript.

We have studied their feedback carefully and have made corrections to our original manuscript, which we hope will meet your approval. The revisions have been highlighted in the text. We answer their comments one by one, as follows:

Reviewer nr 1

Comments to the author:

Name/surname order will be corrected

Comments to the author:

 In order for the review to become even more useful, I would suggest to reorder the Introduction and to shift the second paragraph (line 39-45) behind the sentence in line 56 ending with “potential damage”.

Thank you very much for this important suggestion.

We have changed our Introduction to reflect the suggested order.

Comments to the author:

I further would recommend to include more spectroscopic details about methylene blue in the second section (2. Fluorescence). It would be informative to show an absorption and emission spectrum of methylene blue and to directly specify the wavelength of absorption and emission. And to indicate that it is a hydrophobic molecule.

Thank you very much for this important suggestion.

We have added the following statements:

Methylene blue presents with an absorption peak at 665 nm with a shoulder at 610 nm,  presenting an additional peak at 293 nm38. It has a quantum yield of 0.52. Moreover, the emission spectra of MB is about 686 nm with a stoke shift of 21nm. The clinical properties of MB are limited by its' hydrophobic nature.

We also added a reference:

Selvam S, Sarkar I. Bile salt induced solubilization of methylene blue: Study on methylene blue fluorescence properties and molecular mechanics calculation. J Pharm Anal. 2017;7(1):71-75. doi:10.1016/j.jpha.2016.07.006

Comments to the author:

In line 81 “an excitation peak of about 700 nm, …… an emission of 688nm… “: Please rephrase the sentence such that it is physical correct (stokes shift between absorption and emission). I would suggest to site the following reference J Pharm Anal. 2017 Feb; 7(1): 71–75 or a similar reference

Thank you very much for this important suggestion.

We moved the sentence mentioned in the reviewer's comments (from line 81) a few lines up in our manuscript. Additionally, we added more detailed information about the properties of MB from the suggested paper (J Pharm Anal. 2017)

Comments to the author:

Figure 1. Is the Figure legend missing? It would be nice to highlight in the fluorescence image with an arrow the “ureter during anterior resection”

Thank you very much for this important suggestion.

We have added an arrow marking the fluorescent ureter. The name of the figure is: “Fluorescent visualization of an ureter during rectal cancer anterior resection”

The new figure with an arrow will be uploaded after final revision

On behalf of all the Authors, I would thank the Editor, Editorial Staff and the Reviewers for their important comments and useful suggestions to improve our paper. We appreciate the time and careful thought that went into reviewing our manuscript. We hope that the above changes are to your satisfaction.

Best regards,

Tomasz Cwalinski

Reviewer 2 Report

Tomasz et al review the current usage of Methylene Blue and the application of its fluorescent properties in intraoperative fluorescence imaging. They discuss the potential of Methylene Blue along with its limitations and side effects. The authors have done a good job of reviewing the Methylene Blue NIR imaging field in short. However, there is ample room for improvement in English presentation. There are several places where the review lacks coherence and ideas appear suddenly. Eg. line 44 about patients with liver failure has an unclear antecedent. It is hard to grasp what is the connection of this line to the ideas presented previously. LInes 66-70 need major rework in sentence formation and idea presentation. Lines 69-70  explain fluorophores and fluorescence with a complete lack of necessary details. Lines 80-82 explain the excitation-emission properties of MB in a very convoluted way. It would suffice just to mention the excitation and emission maxima of MB. This paragraph warrants a few lines on the mechanism of action of MB. How does it work at the molecular level? This detail will give a good holistic view to the review from the molecular to the organ level. 

Line 94, dyspnea is misspelled. Line 82-83 points at the less tissue penetration of MB and high autofluorescence of the background tissue.  Lines 150-151 point that after MB administration, autofluorescence was low and SBR was 2.74. Can the authors explain this discrepancy?  

Line 166, units of the extinction coefficient of MB are incorrect. It should be mol/L-1/cm-1. QY of 4.4% compared to what? Line 164, is it umol/L or mmol/L? 

The text of the review is loosely organized and really needs some tightening up. Possibly the limitations and side effects can come at the end of the review after they talk of the 5 main domains.

Author Response

Dear Editor and Editorial staff,

Please find attached the revised version of the manuscript JCM-961668, entitled: “"Methylene Blue - current knowledge, fluorescent properties and its future use."

The paper has been revised in accordance with the Reviewers’ comments. We would like to thank the Reviewers for their important comments and suggestions, which have notably improved the quality of our manuscript.

We have studied their feedback carefully and have made corrections to our original manuscript, which we hope will meet your approval. The revisions have been highlighted in the text. We answer their comments one by one, as follows:

Reviewer nr 2

The text was revised and reviewed again by one of the co-authors who is a native English speaker.

Comments to the author:

There are several places where the review lacks coherence and ideas appear suddenly. Eg. line 44 about patients with liver failure has an unclear antecedent.

Thank you very much for this important suggestion.

We have changed line 44 with the addition of the following sentence: Another field where MB can be used is as an antidote in cyanide poisoning.

Comments to the author:

LInes 66-70 need major rework in sentence formation and idea presentation. Lines 69-70  explain fluorophores and fluorescence with a complete lack of necessary details.

Thank you very much for this important suggestion.

We have moved the following sentence to line 74, allowing for a better understanding of the context of the previous sentence.

Many of the NIR fluorescence systems available on the market can alter the fluorescent signal with an overlaid colour, which facilitates intraoperative orientation.

Comments to the author:

Lines 80-82 explain the excitation-emission properties of MB in a very convoluted way. It would suffice just to mention the excitation and emission maxima of MB.

Thank you very much for this important suggestion.

We have added a new sentence to better summarize this information, incorporating additional feedback provided by reviewer nr 1.

Methylene blue presents with an absorption peak at 665 nm with a shoulder at 610 nm,  presenting an additional peak at 293 nm38. It has a quantum yield of 0.52. Moreover, the emission spectra of MB is about 686 nm with a stoke shift of 21nm. The clinical properties of MB are limited by its' hydrophobic nature.

Comments to the author:

This paragraph warrants a few lines on the mechanism of action of MB. How does it work at the molecular level? This detail will give a good holistic view to the review from the molecular to the organ level. 

Thank you very much for this important suggestion.

This comment raises a very good point. We have added a short detail to our manuscript about MB's function at the molecular level. However, we intentionally did not focus on it's effects at the molecular level, as this is an additional function and could be reviewed on its' own entirely.

We have added the following sentence:

The main effect of methylene blue on molecular level is through modulation of the cGMP pathway. Additionally, multiple cellular and molecular targets of the MB compound have been identified. Many studies describe in detail the mechanisms of MB at the molecular level. 29,30

Oz M, Lorke DE, Hasan M, Petroianu GA. Cellular and molecular actions of Methylene Blue in the nervous system. Med Res Rev. 2011;31(1):93-117. doi:10.1002/med.20177

Nedu ME, Tertis M, Cristea C, Georgescu AV. Comparative Study Regarding the Properties of Methylene Blue and Proflavine and Their Optimal Concentrations for In Vitro and In Vivo Applications. Diagnostics (Basel). 2020 Apr 15;10(4):223. doi: 10.3390/diagnostics10040223

Comments to the author

Line 94, dyspnea is misspelled

Thank you for pointing this out. This error has been corrected.

Comments to the author:

Line 82-83 points at the less tissue penetration of MB and high autofluorescence of the background tissue.

Thank you for this suggestion. We have already changed these lines with the abovementioned corrections.

Comments to the author:

Lines 150-151 point that after MB administration, autofluorescence was low and SBR was 2.74. Can the authors explain this discrepancy?  

Thank you very much for this important suggestion.

The authors of the manuscript reviewed, stated that the SBR they obtained was 2.74.  In our experience, the SBR in regards to MB could be this high. Theoretically, the SBR should be lower because of this phenomenon, but it seems that fluorescence of this compound is high enough to be easily visualized.   

Comments to the author:

Line 166, units of the extinction coefficient of MB are incorrect.

Thank you for pointing this out. We have changed the extinction coefficient units to:  69100 mol/L-1 cm-1

Comments to the author:

QY of 4.4% compared to what?

Thank you for bringing this to our attention. 

Upon review, we deemed this unnecessary information and have deleted this part.

Comments to the author:

Line 164, is it umol/L or mmol/L? 

Thank you for pointing this out.

Indeed, it was a typing error. We have corrected it to 20 umol/L.

Comments to the author:

The text of the review is loosely organized and really needs some tightening up. Possibly the limitations and side effects can come at the end of the review after they talk of the 5 main domains.

Thank you very much for this important suggestion.

We have reviewed our manuscript again in light of this feedback and have reorganized the proposed sections for better flow and structure. We have added subheadings and additional points when necessary as well.

On behalf of all the Authors, I would thank the Editor, Editorial Staff and the Reviewers for their important comments and useful suggestions to improve our paper. We appreciate the time and careful thought that went into reviewing our manuscript. We hope that the above changes are to your satisfaction.

Best regards,

Tomasz Cwalinski
